# Theoretical Study of Cyanidin-Resveratrol Copigmentation by the Functional Density Theory

**DOI:** 10.3390/molecules29092064

**Published:** 2024-04-30

**Authors:** Breyson Yaranga Chávez, José L. Paz, Lenin A. Gonzalez-Paz, Ysaias J. Alvarado, Julio Santiago Contreras, Marcos A. Loroño-González

**Affiliations:** 1Departamento Académico de Fisicoquímica, Facultad de Química e Ingeniería Química, Universidad Nacional Mayor de San Marcos, Lima 15081, Peru; 2Departamento Académico de Química Inorgánica, Facultad de Química e Ingeniería Química, Universidad Nacional Mayor de San Marcos, Lima 15081, Peru; 3Instituto Venezolano de Investigaciones Científicas (IVIC), Centro de Biomedicina Molecular (CBM), Laboratorio de Biocomputación (LB), Maracaibo 4001, Zulia, República Bolivariana de Venezuela; 4Instituto Venezolano de Investigaciones Científicas (IVIC), Centro de Biomedicina Molecular (CBM), Laboratorio de Biofísica Teórica y Experimental (LQBTE), Maracaibo 4001, Zulia, República Bolivariana de Venezuela; 5Departamento Académico de Química Orgánica, Facultad de Química e Ingeniería Química, Universidad Nacional Mayor de San Marcos, Lima 15081, Peru

**Keywords:** cyanidin, resveratrol, copigmentation, DFT, non-covalent interaction, isosurfaces

## Abstract

Anthocyanins are colored water-soluble plant pigments. Upon consumption, anthocyanins are quickly absorbed and can penetrate the blood–brain barrier (BBB). Research based on population studies suggests that including anthocyanin-rich sources in the diet lowers the risk of neurodegenerative diseases. The copigmentation caused by copigments is considered an effective way to stabilize anthocyanins against adverse environmental conditions. This is attributed to the covalent and noncovalent interactions between colored forms of anthocyanins (flavylium ions and quinoidal bases) and colorless or pale-yellow organic molecules (copigments). The present work carried out a theoretical study of the copigmentation process between cyanidin and resveratrol (CINRES). We used three levels of density functional theory: M06-2x/6-31g+(d,p) (d3bj); ωB97X-D/6-31+(d,p); APFD/6-31+(d,p), implemented in the Gaussian16W package. In a vacuum, the CINRES was found at a copigmentation distance of 3.54 Å between cyanidin and resveratrol. In water, a binding free energy ∆G was calculated, rendering −3.31, −1.68, and −6.91 kcal/mol, at M06-2x/6-31g+(d,p) (d3bj), ωB97X-D/6-31+(d,p), and APFD/6-31+(d,p) levels of theory, respectively. A time-dependent density functional theory (TD-DFT) was used to calculate the UV spectra of the complexes and then compared to its parent molecules, resulting in a lower energy gap at forming complexes. Excited states’ properties were analyzed with the ωB97X-D functional. Finally, Shannon aromaticity indices were calculated and isosurfaces of non-covalent interactions were evaluated.

## 1. Introduction

Anthocyanins belong to a subgroup of flavonoids of polyphenols and correspond to a wide range of colors of various plants. Anthocyanin is in the form of glycoside, and the aglycone is anthocyanidin, a flavylium cation. The extended conjugation is responsible for its red pigmentation. The skeleton of the flavylium ion (2-phenylchromenylium) is shown in Figure 1 [1,2].

Due to their chemical instability, free-state anthocyanins rarely exist in nature and are easily degraded during processing and storage [2,3,4,5]. There are different methods by which anthocyanins can be stabilized [5,6,7]. Copigmentation, for example, is a strategy that has attracted considerable attention from scientists, as this process has been commonly observed in many anthocyanins acylated with hydroxycinnamic aromatic derivatives. It involves non-covalent interactions (primarily π-π stacking) between the anthocyanidin backbone and its covalently bound acylation fractions, which can protect the chromophore from hydration and result in a bathochromic change of the visible band. Intramolecular copigmentation resulting from multiple aromatic groups in polyacylated anthocyanins can occur both above and below the 2-phenylchromenylium backbone, forming a sandwich-like configuration. Hydrogen bonding, weak intermolecular forces, interactions, especially those between polarizable orbitals of aromatic rings, and hydrophobic effects promote an anthocyanin–copigment association. These intermolecular interactions between the polarizable flat nuclei of anthocyanin constitute the copigmentation force. The non-covalent interactions may protect the anthocyanin molecule from nucleophilic attacks by water [8,9,10].

Alzheimer’s disease (AD) is the most common cause of dementia and an increasingly common cause of morbidity and mortality in the elderly. In many studies, it has been discovered that resveratrol (RV) has many potential health benefits, like antioxidant, cardioprotective, neurological, anti-inflammatory, antiplatelet, blood glucose-lowering, and anticancer activities. Recently published literature has shown that RV defends against some neurodegenerative diseases, like AD and obesity, as well as being effective in treating osteoporosis in postmenopausal women by assuring low risk against breast cancer [11]. In the case of AD, resveratrol promotes the non-amyloidogenic cleavage of the amyloid precursor protein, enhances clearance of amyloid beta-peptides, and reduces neuronal damage. Despite the effort spent trying to understand the mechanisms by which resveratrol functions, the research work in this field is still incomplete [12].

Quantum mechanics DFT screening can provide microscopic interactive conformation, UV spectrum, and binding free energy of copigmentation systems. However, the use of DFT functionals should include long-range electron–electron correlation effects. Dispersion-corrected functionals address this problem, making it possible to accurately model the interactions that enable copigmentation [13,14,15,16,17,18,19,20,21,22,23,24]. Long-range dispersion and/or separation corrections such as gd3bj (a damping function) [25,26] allow managing the transition from short to long distances to avoid divergence problems. Advanced theoretical calculations can support experimental data in identifying individual contributions to the stability of the copigmentation complex (e.g., hydrophobic effect, dispersion forces, π−π stacking, exchange term, and hydrogen bonding). By understanding these interactions, we can rationalize their spectral consequences (i.e., bathochromic and hyperchromic changes).

The works published on resveratrol in Alzheimer’s disease treatment and copigmentation studies motivated this theoretical investigation. Better understanding of the intermolecular interaction, properties, and stability of the copigmentation complexes will contribute to the investigation of the potential use of cyanidin-resveratrol (CINRES) complex in Alzheimer’s disease treatment. On this basis, we carried out a computational study of the CINRES complex by addressing possible non-covalent interactions that could be involved in its formation. To do so, several DFT functionals are proposed: the ωB97X-D, M06-2X, and APFD. The first is known as a range-separated functional, which can capture both short-range and long-range interactions. It includes empirical dispersion and is highly recommended for theoretical studies of copigmentation. Similarly, the M06-2X functional is a high non-locality functional with twice the amount of non-local exchange (2X) and is parameterized only for nonmetals [27]. On the other hand, the scatter-corrected APFD functional provides near-thermochemical accuracy comparable even to composite methods, avoiding the long-range attractive or repulsive false interactions found in most density functional theory models [28]. At the time of this study, the CINRES complex has not been proposed as a possible treatment for neurodegenerative diseases, as an inhibitor of the enzyme acetylcholinesterase.

## 2. Results

### 2.1. Quantum Molecular Calculations (QMCs)

All quantum calculations were performed through the GaussView interface of the Gaussian16 program. The functional M06-2x, along with a 3-21G* level base assembly, was used as a filter to obtain its binding energies until a pattern was obtained and applied to the rest of the complexes. Figure 1 shows the most stable structure which converged when both molecules (pigment and copigment) were completely coplanar with an intermolecular distance of about 4.0 Å.

The possible conformers were obtained using the Genmer module in Molclus software (version 1.9.9.4). Initially, the program generated nine random structures, which were later optimized by combining semi-empirical and quantum methods, i.e., PM6 and M06-2X/3-21g* (gd3bj). All calculations generated structures with zero negative frequencies, identified as minimum energy structures. Figure 2 contains nine of the conformations generated, selected according to the Boltzmann distribution factor. Not all of the nine structures were successfully optimized, i.e., many of them had convergence failures. All conformations were given by the Molclus program. Once the geometries of all complexes were optimized, using the theory level M06-2X/3-21* (gd3bj), the Boltzmann distribution factor was applied, as an indication of the stability of the conformer to be used. The partition functions were computed through GaussView, in the session where the spectroscopic frequencies were calculated.

### 2.2. Calculation of the Binding Energy

The energy of binding (or association), ∆*E_biding_*, for two weakly interacting *M* and *N* (sub)systems has a profound effect on the supramolecular self-assembly of copigments. In this work, this energy was determined as follows:(1)$${∆E}_{biding}=E_{MN\;complex}^{BSSE}-E_{M}-E_{N}$$

Table 1 shows the calculation at different levels of theory; Table 2 shows the calculation of the binding energy using Formula (1). Calculated values do not show important variations with different DFT functionals, and basis effect with and without BSSE corrections.

### 2.3. Binding Free Energy Calculation

The theoretical Gibbs free energy change was calculated as:(2)$$∆G_{binding}=∆G_{g}+E_{P\ldots CoP}^{solvent}+E_{P}^{gas}+E_{coP}^{gas}-\left(E_{P}^{solvent}+E_{coP}^{solvent}+E_{P\ldots CoP}^{gas}\right)$$
where $∆G_{g}$ means the free energy change in a vacuum and the rest of the terms represent the energies as calculated in Table 1. Table 2 shows all the calculations performed for each level of theory in both a vacuum and water as a solvent. Table 3, Table 4 and Table 5 show the thermodynamic results at 298 K, both for cyanidin plus resveratrol individually and in complex form.

Finally, Table 6 shows the calculation of ∆*G_binding_*, where the effect of the solvent on the binding free energy in the pigment–copigment complex is clear and, for the entire level of theory applied, the following values in the vacuum were found: −13.65; −11.39; −16.89 kcal/mol. In said table, we can find in bold (10.34; 9.71; 9.98) how the effects of the water solvent are mitigated by positive values. However, it is not enough to prevent the formation of the complex in water, since, in all cases, the resulting free energy values were all negatives.

### 2.4. Topological Analysis of the Electron Density

We carried out calculations of non-covalent interactions that could be present in the CENRES complex using VDM (version 1.9.4) and MULTIWFN (version 3.8) software. Non-covalent interaction (NCI) index has become a versatile tool for analyzing the presence and strength of localized and delocalized interactions in systems of interest, used in diverse areas such as biochemistry, reactivity, and solid state. In this case, DFT electron densities were calculated using the functionals (M06-2x/6-31g+(d,p) (gd3bj), ωB97X-D/6-31g+(d,p), and APFD/6-31g+(d,p)). This procedure involved the use of the reduced density gradient methodology. Once obtained, large green areas revealed a pattern of non-covalent interactions. The red areas, indicative of strong internal repulsions in the center of the aromatic centers, were only present within the rings. On the right side in Figure 3, the gradient graphs are displayed, representing the reduced density as a function of the sign of the second eigenvalue of the Hessians, multiplied by the density, shown here in colors to highlight the present interactions. From the figure, it is evident that no marked differences were found when using different functionals.

### 2.5. UV Spectra

The classic density functional theory (DFT) formalism cannot treat time-dependent (TD) problems nor describe excited electronic states. However, information about electronic excited states may be obtained through the linear response (LR) theory formalism. In this section, to obtain UV-TD-SCF spectra, we only took into account the ωB97X-D functional and basis set 6-31g+(d,p). We started with the cyanidin and resveratrol, and then moved on to calculate the theoretical spectra of the double and triple complexes. In the case of triple copigmentation, the resveratrol sandwiches at cyanidin. In each Table, the strongest transition between HOMO-n → LUMO-m orbitals, and n and m are orbital indices are marked according to the oscillator strength. For example, for the case of the cyanidin, the HOMO-74 to LUMO-75 orbitals will be involved in the UV transition.

#### 2.5.1. Cyanidin UV Analysis

Orbital calculations show only three transitions with sufficient transition force to be observed, the strongest one resulting from UV frequency 474.91 nm, corresponding to orbitals 74 (HOMO) and 75 (LUMO), its transition occurring with an oscillator strength of 0.5722 (Table 7). The percentage of the coefficient would be (0.67064 × 0.67064) × 2 × 100, equal to 89.95%. Figure 4 shows the intensity of the three transitions, while Figure 5 shows the type of orbitals involved.

#### 2.5.2. Resveratrol UV Analysis

As in the case of cyanidin, resveratrol showed only three strong signals. The strongest signal corresponded to the 343.56 nm transition, the molecular orbitals, shown in Table 8. With resveratrol, the force magnitude of this oscillator was more than 94%, involving the orbitals 60 and 61. In this case, the diagrams HOMO-1, HOMO, LUMO, and LUMO + 1 were plotted to give a pictorial explanation of the energy gaps (Eg) and their distributions. From Figure 6, Eg is 4.60 eV is the highest, but the strongest occurs at 3.60, which is relatively less reactive than the 2.60 eV cyanidin alone; therefore, excitation from HOMO to LUMO is not very feasible. Figure 7 shows its UV spectrum.

#### 2.5.3. Complex: Cyanidin–Resveratrol UV Analysis

Here the signal is greatly diminished, the strongest having a strength of 27% at 468.83 nm, slightly lower than that of cyanidin alone, and there is no evidence of the resveratrol signal (Table 9). If we look at the HOMO and LUMO orbitals, Figure 8 shows the UV spectrum of the complex and Figure 9 shows the orbitals. We observe resveratrol giving up its electrons to cyanidin, making it react.

It is interesting to point out that all transitions of the doublet complex arrive at a single state, the LUMO. This state possesses all the electrons, which makes it possible to see a single signal, as if there were only cyanidin (Figure 10).

#### 2.5.4. Complex: Resveratrol–Cyanidin–Resveratrol UV Analysis

These results generated a behavior very similar to those from the case of the double complex. Surprisingly, the cyanidin in the LUMO state takes all the interacting electrons, and the signal is practically unified. Table 10 shows all allowed transitions and oscillation strengths.

Normally, if the bandwidth is below 2.60 eV, a molecule can be considered reactive. This tendency is evidenced by having a triple complex. Here, the band gap drops to 2.57 eV, and the signal becomes weaker (see Figure 11 and Figure 12). Figure 13 shows the UV spectrum of the triple complex.

As a final comment on this UV section, it is important to remember that the energy difference between the LUMO and HOMO energies reflects the chemical reactivity and kinetic stability of any molecule or complex. So, as this difference is small, the flow of electrons to the lower energy state is higher. This means the molecule is polarizable and generally associated with a high chemical reactivity, and low kinetic stability; it is labelled as soft. With that in mind, if we analyze all four UV-TD-SCF calculations for our systems, cyanidin, resveratrol, cyanidin–resveratrol, and resveratrol–cyanidin–resveratrol, from Table 11, by complexing the resveratrol with cyanidin into different species, it will make them a more polarizable and reactive system, thus kinetically less stable.

The results show the most important theoretical results for electronic transitions, the corresponding vertical absorption wavelengths (λcalc) and oscillator strengths (f) of the four analyzed molecules calculated particularly with the DFT: ωB97X-D/6-31+G(d,p) methodology. It was observed that the complex absorbance intensity is significantly lower. However, such relative differences in absorbances were not quantified or even theoretically explained; we only know that the functional ωB97X-D, as well as including dispersion factors, reaches a 22% HF exchange in the short range and a range separation parameter of 0.2.

### 2.6. Topological Analysis: Use of IGM Software

#### IBSI (Intrinsic Bond Strength Index)

The IBSI (Intrinsic Bond Strength Index) is a very efficient scoring function for internally probing the strength of a given pair of atoms in a molecule, and it is closely related to the local binding stretch force constant. It derives from the integration of IGM-δg^pair^ obtained for a given pair of atoms, using the new link descriptor δg^pair^. The IGM software version used by this calculation was 3.08. This was calculated as given in Equation (3):(3)$$IBSI=\frac{\int_{V}\frac{\delta g^{pair}}{d^{2}}dV}{\int_{V}\frac{\delta g^{H_{2}}}{d_{H_{2}}^{2}}dV}$$

*δg^pair^*, as we explained, is the descriptor that quantifies the electron density (ED) countergradiance between A and B, the d-coordinate being the internuclear distance between them. IBSI has been normalized to 1 for H2 at the theory level M06-2X/6-31G(d,p). The calculations are presented in visual form in Figure 14.

Table 12 shows the IGM calculation for the hydrogen bond interaction, clearly visible through the isosurfaces, with a value of 0.048 for the H26-O46 system, i.e., hydrogen 26 with oxygen 46. The migration trend of electrons is observed to be H26 → O46. In the case of the other interactions, it is much weaker, at 0.004. The results were consistent for the ωB97X-D, APFD, and M06-2X functionals. In all cases, the same base set 6-31+g(d,p) was used (Table 13 and Table 14).

### 2.7. Aromaticity Index of Shannon (SA)

The entropy method is used to determine the aromaticity of each molecule, as monomers and in the complexes. A change in these magnitudes gives us a measure of the loss of aromaticity due to electron density transfer during the formation of the complexes. For this purpose, we used the MultiWfn program and evaluated the critical points to estimate the Shannon aromaticity index. To determine the nature of our system, we use the range of 0.003 < SA < 0.005 as the boundary between aromatic and antiaromatic systems.

Table 15 shows how the aromaticity index is affected; the smaller the value the better the aromaticity index and, as the value increases, the aromaticity index is harmed. In the formation of the complex, it is seen how the aromaticity index increases; this indicates that there is electron transfer between the molecules of the complex. Also, in this table, we can observe that when cyanidin is in free form, it has an aromaticity index of 0.00067, but if it is complexed, it shows a clear increase in the index (see red lines in Table 15). The same trend happens with resveratrol, which goes from 0.00122 to 0.00170 on average in its SA. This loss of aromaticity could be associated with a charge transfer between cyanidin and resveratrol, and it may be associated with the decrease in absorption bands for the UV theoretical spectra.

## 3. Theoretical Details

To find the best initial structure of the copigmentation complexes between flavylium ions and copigments, the MOLCLUS software was used [29]. All complexes obtained by MOLCLUS were pre-optimized by a semi-empirical PM6 model. The geometries of the three complexes were optimized using DFT and the ωB97X-D, M06-2x, and APFD functionals, combined with base sets 3–21 g, 6-31G(d,p), and 6-31+G(d,p) [26,30]. Frequency calculations ensured that each geometry corresponded to minimum energy conformers by ensuring that all their vibrational frequency values were positive and within a convergence scale of 1 × 10^−8^ a.u. of energy. This process was achieved with the help of the Gaussian16 computational package [31].

The intermolecular interaction energies were calculated as follows:(4)$$∆E_{biding}=E_{complex}-\sum_{i}^{free\;molecules}E_{i},$$
where $E_{complex}$ denotes the energy of the complex, and the sum goes over the two free molecules. However, as the atoms of interacting molecules approach one another, there exists an error that normally occurs in quantum chemistry calculations using finite basis sets called Basis Set Superposition Error (BSSE); this error can be calculated using the traditional counterweight method:(5)$$BSSE=\left[E_{AB}^{AB}(A)-E_{A}^{AB}(A)\right]+\left[E_{AB}^{AB}(B)-E_{B}^{AB}(B)\right],$$
where $E_{AB}^{AB}(A)$ y $E_{AB}^{AB}(B)$ are the energies of two given free molecules *A* and *B*, respectively, as obtained in complex geometry *AB* with the basis set *AB*. $E_{A}^{AB}(A$) and $E_{B}^{AB}(B$) are the energies of *A* and *B*, respectively, obtained in the complex geometry *AB* with the basis sets *A* and *B* [32]. In the end, this error can be subtracted from the original calculation of $∆E_{biding}$.

### 3.1. Calculation of the Free Energy of the Complex

The theoretical change in Gibbs free energy from gas ($∆G_{theo}^{gas}$) for the copigmentation was calculated using the following formula:(6)$$∆G_{theo}^{gas}=G_{complex}-G_{pigment}-G_{copigment}$$
where $G_{complex}$, $G_{pigment}$, and $G_{copigment}$ represent the free Gibbs energy of the copigmentation complex, pigment, and copigment, respectively.

A realistic determination of the $∆G_{theo}^{gas}$ includes a solvent effect centered on Gibbs’s association energies (biding), $∆G_{biding}$. According to the first principle of thermodynamics, the Gibbs binding energy is negative for complexes that form spontaneously; the lower the $∆G_{biding}$, the more stable the complex would be. For the calculation of the $∆G_{biding}$, the thermodynamic cycle of Figure 2 was used. This allowed the process to be divided into two parts, one corresponding to the formation of complexes in the gas phase and the other to the transfer of the process from the gas phase to a solvent, which in this case is water.

The Gibbs association energy is then estimated using the following formula [33]:(7)$$∆G_{biding}=\langle {∆H}_{MM}\rangle -T\langle {∆S}_{MM}\rangle +\langle {∆G}_{solv}\rangle ,$$
where ${∆H}_{MM}$ is enthalpy in the gas phase; ${∆S}_{MM}$ is the entropy of the solute in the gas phase; and ${∆G}_{solv}$ is the Gibbs solvation energy, all averaged along the MD trajectory.

In this paper, though, we proposed a different approach to calculating $∆G_{biding}$:(8)$$∆G_{biding}=∆G_{g}+E_{P\ldots CoP}^{solvent}+E_{P}^{gas}+E_{coP}^{gas}-(E_{P}^{solvent}+E_{coP}^{solvent}+E_{P\ldots CoP}^{gas}).$$

With this approach, we only correct $∆G_{g}$ values using an energy factor due to the presence of the solvent without relaxing complex structures in the solvent, reducing the computational cost.

### 3.2. Topological Studies through the Density Matrix

A topological study based on the density matrix of quantum mechanics will help us understand hydrogen bonds, π-π interactions, and other critical interaction points evaluated using atoms in molecule theory (AIM), including bond critical points (BCPs), ring critical points (RCPs) and cage critical points (CCPs). To further investigate non-covalent interactions, the isosurfaces of the reduced density gradient (RDG) were plotted using Multiwfn [34], IGMPLOT [35,36,37,38,39,40], and NCIPLOT (version 4.2) [41] software’s. Conformations and non-covalent interactions were calculated using VMD software [42]. Non-covalent interactions (NCIs) determine how molecules approach each other and eventually bundle together [43,44,45]. NCIs cover a wide range of non-covalent interaction types, such as hydrogen and halogen bonds, CH-π and π-π interactions, and various bonding (or anti-bonding) forces, e.g., dispersion, electrostatics, and Pauli’s exclusion principle. Decoding the intermolecular interactions NCI in the creation of supramolecular ensembles represents a critical step in advancing structural prediction [46], molecular reactivity, and drug–receptor docking processes.

An NCI descriptor, introduced by Johnson et al. [47], is used to determine non-covalent interactions. It is a dimensionless quantity, based on the reduced density gradient (RDG), *s*(*r*), and is a measure of how the electron density (ED), ρ(*r*), deflects locally from that of a homogeneous electron gas, which has, by definition, ***s***(*r*) everywhere. The RDG is a fundamental quantity in the DFT calculations. The minimum value of ***s***(*r*) is zero, regardless of the type of ED used, and it occurs whenever the ED gradient disappears at critical points in ρ(*r*), as seen in bond critical points (BCPs) and ring or cage critical points. The ***s***(*r*) can be calculated by
(9)$$s\left(r\right)=\frac{1}{2\;\sqrt[{3}]{3\pi^{2}}}\frac{\left|\nabla \rho (r)\right|}{\sqrt[{3}]{{\rho (r)}^{4}}}.$$

The RDG assumes large values in regions far from the various nuclei of a system, where the total ED is decaying to zero exponentially and the term *ρ*(*r*)^4/3^ approaches zero faster than $\left|\nabla \rho (r)\right|$. According to Johnson, the sign of the second highest eigenvalue λ_2_ (λ_1_ ≤ λ_2_ ≤ λ_3_) of the ED burlap matrix at each point on the isosurfaces would distinguish (locally) between attractive (λ_2_ < 0) or repulsive interactions (λ_2_ > 0). Mapping the value “*ρ*(*r*)sign (λ_2_)” on each RDG isosurface can qualitatively reveal both the nature and strength of the interactions.

### 3.3. Calculation of the Aromaticity of the Complex

Shannon’s aromaticity (SA) index was used to calculate this property [48], which determined the magnitude of the electron density at each critical point of the studied structure, and then the following procedure was applied:First, find the probability of localized electron density at critical points:
(10)$$p_{i}=\frac{\rho_{i}\left(r_{c}\right)}{\sum_{i=1}^{N}\rho_{i}{(r}_{c})}$$

Then, calculate the entropy of local information:


(11)
$$S_{i}\left(r_{c}\right)=-p_{i}\left(r_{c}\right)lnp_{i}{(r}_{c})$$


Sum up all the contributions of local information:


(12)
$$S_{t}(r_{c})=\sum_{i=1}^{N}S_{i}{(r}_{c})$$


Calculate the maximum allowable entropy as:


(13)
$$S_{max}{(r}_{c})=\ln(N)$$


Finally, the aromaticity index is calculated as:


(14)
$$SA=S_{max}-S_{t}$$


An important point to consider here is that the number of suitable critical points in the calculation of *SA* is the main problem of the proposed procedure above. Once we have defined them for each pair of bonding or non-bonding atoms, the *SA* can be used as a measure of the spatial localization of the electron density and shows the extent of aromaticity or antiaromaticity of the system. In general, the system is more aromatic as the *SA* index goes to zero. The range 0.003 < *SA* < 0.005 is obtained as the boundary between aromatic and antiaromatic systems [48].

## 4. Conclusions

After a process of geometric optimization of a complex set between cyanidin and resveratrol, a stable cyanidin–resveratrol complex was found. Calculations were performed to obtain energetic, thermodynamic, structural, and aromatic properties. These parameters and properties help understand the dynamics of their interactions in the different forms of aggregation. These complexes were modeled using different functionals M06-2X, ωB97X-D, and APFD, and the basis set 6-31+(d,p). For the quantum calculations, we used Gaussian16, IGMPLOT, and MULTIWFN, as well as the VMD and the GaussianView visualizers. This work serves as a basis for further development of docking and molecular dynamics analysis of the interaction between the cyanidin–resveratrol complex and the acetylcholinesterase enzyme.

## Data Availability

Data are contained within the article.

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
