# Peer review of "Theoretical Study of Cyanidin-Resveratrol Copigmentation by the Functional Density Theory"

_molecules, 2024, doi:10.3390/molecules29092064_

Round 1

Reviewer 1 Report

Comments and Suggestions for Authors

Comments on molecules-2922418:

This is a low-quality work with many problematic points in experiment design, presentation and reasoning. The authors lack a solid background in relevant fields (e.g., quantum chemistry and statistical mechanics). The quality and scientific rigor of the work seem questionable, and a huge number of writing problems could be identified. As a result, I cannot recommend its publication.

Many formatting and writing problems exist throughout the paper. For example, ‘biding’ in the first paragraph of section 3.2 is obviously incorrect. Scheme 4 is incomplete. The structures of the cis- and trans- conformations of the same molecule, resveratrol, are inconsistent. The left one (trans) has a -OH substitution and two -CH3 groups, while the right one (cis) has three -CH3 groups. Such problems suggest the careless writing style when preparing the manuscript.

More importantly, some of presentation problems concern scientific contents. What is this ‘free energy binding energy’ mentioned in abstract? Such a term does not exist in thermodynamics. Similar problems exist on page 5, where the authors write ‘free Gibbs energy’, a wired statement. These problems indicate that the authors lack a solid background in statistical mechanics (thermodynamics).

Concerning the functional name, what is this WB97XD functional? The official name should be ωB97X-D. M062X should be M06-2X. While in many cases such as programming, aliases are employed, in a scientific publication the exact name of DFT levels should be used.

Another problem in DFT calculations concerns the level of theory for the calculation of energetic data. With literally no conformational sampling, the computational costs of these single-point calculations are rather low. Considering this fact, normally in a QM-calculation-only investigation, practitioners are using higher-level selections to obtain more accurate energetics, such as using a def2-TZVP-level or higher basis sets. The current 6-31G-level Pople selection is absolutely unacceptable and unreliable. Further problems exist in other details of QM calculations. For instance, the authors are investigating the non-covalent interactions of multiple molecules and the real-world environment is absolutely not vacuo. As a result, some treatment of solvents should be added, such as using some implicit-solvent models to represent water. However, these details are simply forgotten, which obviously makes the computational outcome relevant only to gas-phase interactions that are not representative of water solution.

The motivation of this work seems rather weak and irrelevant to any real-world applications. The authors mentioned AD, amyloids and many other keywords in their manuscript, but only investigated the non-covalent interactions between small drug-like molecules. In practical situations, these small molecules are interacting with proteins, which are much more complex than any systems that the current investigation focuses on.

Too many remaining problems exist but would not be unmentioned. The reviewer has limited time in the volunteering peer review process. In any case, such a paper without redesign of the experiment, the addition of additional scientific findings, and thorough rewriting cannot be considered for publication.

Comments on the Quality of English Language

Too many writing and language problems. 

Author Response

Please read my letter

Thanks

Marcos Loroño

Reviewer 2 Report

Comments and Suggestions for Authors

See enclosed file

Comments on the Quality of English Language

Moderate English revision should be performed

Round 2

Reviewer 1 Report

Comments and Suggestions for Authors

Although the authors have improved their manuscript in this round, I still have concerns about novelty and scientific rigour. While the authors seem unwilling to enrich the scientific findings by including additional investigation, I would suggest to at least redo their calculations at a higher level (e.g., basis set def2-TZVP) to ensure the reliability of the energetics. 

Comments on the Quality of English Language

Improvable.

Author Response

Thank you for your comments, we appreciate your concerns about novelty and scientific rigour. Regarding your observation, the Cyanidin-Resveratrol (CR) complex was the result of the use of six types of anthocyanidins such as pelargonidin, petunidin, peonidin, delphinidin, cyanidin and malvidina, aside from the Resveratrol. First, we performed (M06-2X)-DFT calculations combining anthocyanins and Resveratrol in pairs. From those calculations, the CR complex possessed the highest interaction energy in modulus. In a second phase, the CR or Resveratrol-Cyanidin-Resveratrol (RCR) complexes were used along with the functionals: wB97X-D, APFD and M06-2x with D3-BJ (Grimme's dispersion correction with Becke-Johnson Damping), plus the same 6-31+(d,p) basis set. Our purpose with this research was to consider non-covalent interactions of the aforementioned sets with the Resveratrol, and, at this point, never considered using a def2-TZVP basis set. That surely would have been quite interesting to see through, we will likely take this recommendation in future projects.

Your proposal was estimated by our group in order to give completeness to the energy studies. For other researches we are using the methodology proposed by you and the calculation times in our equipment are very high.  As indicated in our title, we consider in this manuscript the part I of the research, being sure and necessary to explore with basis set def2-TZVP to give a higher level of calculation in molecular systems like the mentioned and others of singular interest.  We are very sorry at this time not to provide the requested result, in view of the delay that this would imply in the generation of the results, analysis in incorporation in the present manuscript.

Finally, we thank for your time and dedication in reading and analyzing the manuscript, as well as for their suggestions for improving its content. We have taken advantage of this review to incorporate in the manuscript small corrections in the spelling, typography and standardization of the references adjusted to the requirements of the journal, without this implying a modification of the sense or syntax of what has already been evaluated.

Sincerely,

Marcos Loroño

Reviewer 2 Report

Comments and Suggestions for Authors

The authors answered all the referee's criticisms and modified consequently the original manuscript.

Now the manuscript has been improved and can be published as it is.

Author Response

We appreciate your answer.

we thank for your time and dedication in reading and analyzing the manuscript, as well as for their suggestions for improving its content. We have taken advantage of this review to incorporate in the manuscript small corrections in the spelling, typography and standardization of the references adjusted to the requirements of the journal, without this implying a modification of the sense or syntax of what has already been evaluated.

Sincerely,

Marcos Loroño